Comprehensive analysis of the correlations of S100B with hypoxia response and immune infiltration in hepatocellular carcinoma

Yan Jia 1 2
Huang Ya jun 2
Huang Qing yu 2
Liu Peng Xia 2
Wang Chang Shan 1 2 changshanwang@imu.edu.cn
1 Department of Bioscience, State Key Laboratory of Reproductive Regulation and Breeding of Grassland Livestock, Inner Mongolia University , Hohhot , China
2 Department of Bioscience, Inner Mongolia University , Hohhot , China
Uversky Vladimir
Electronic publication date: 2022 Mar 29
Publication date: 2022
Volume: 10
Electronic Location ID: e13201
Received 2021 Dec 6; Accepted 2022 Mar 9
Copyright: © 2022 Yan et al.
Copyright year: 2022
Copyright holder: Yan et al.
License: This is an open access article distributed under the terms of the Creative Commons Attribution License, which permits unrestricted use, distribution, reproduction and adaptation in any medium and for any purpose provided that it is properly attributed. For attribution, the original author(s), title, publication source (PeerJ) and either DOI or URL of the article must be cited.
License URL: https://creativecommons.org/licenses/by/4.0/

Keywords: Hepatocellular carcinoma, S100B, Hypoxia, HIF-1α, Immune infiltration

Funding: National Natural Science Foundation of China 81660024, 32060156 Natural Science Foundation of Inner Mongolia 2020MS08096, 2020MS03007 This work was supported by the National Natural Science Foundation of China (CN) (No. 81660024 and No. 32060156), and the Natural Science Foundation of Inner Mongolia (No. 2020MS08096 and No. 2020MS03007). The funders had no role in study design, data collection and analysis, decision to publish, or preparation of the manuscript.

==============================
S100B has been found to be dysregulated in many cancers including hepatocellular carcinoma (HCC). However, the functions of S100B and its underlying mechanisms in HCC remain poorly understood, especially in the tumor microenvironment. In this study, functions enrichment analysis indicated that S100B expression was correlated with hypoxia and immune responses. We found that hypoxia could induce S100B expression in an HIF-1α-dependent manner in HepG2 cells. Luciferase reporter and ChIP-qRCR assays demonstrated that HIF-1α regulates S100B transcription by directly binding to hypoxia-response elements (HREs) of the S100B promoter. Functionally, knockdown of S100B reduces hypoxia-induced HepG2 cell invasion and migration. Furthermore, GSVA enrichment results displayed that S100B and its co-expressed genes were positively correlated with EMT pathway in HCC. Additionally, GO/KEGG cluster analysis results indicated that co-expressed genes of S100B were involved in biological processes of immune response and multiple tumor immune-related signaling pathways in HCC. S100B expression was positively correlated with multiple immune cells tumor infiltration and associated with chemokines/chemokine receptors and immune checkpoint genes. Moreover, S100B is predominantly expressed in immune cells, especially NK (Natural Killer) cell. In addition, the hub genes of S100B co-expression and hypoxia response in HepG2 cell were also associated with immune cells infiltration in HCC. Taken together, these findings provide a new insight into the complex networks between hypoxia response and immune cells infiltration in tumor microenvironment of liver cancer. S100B maybe serve as a novel target for future HCC therapies.

Introduction

Liver cancer is one of the most common cancers that seriously threatens human health and life (Asrani et al., 2019; Siegel, Miller & Jemal, 2020). Hepatocellular carcinoma (HCC) is the most common primary liver cancer, and accounts for approximately 90% of liver cancers (Asrani et al., 2019). It is the fourth-leading cause of cancer death worldwide (Asrani et al., 2019). Despite advances in therapeutic methods, the survival rate of liver cancer patients is still low, due to the development of recurrence and metastasis (Siegel, Miller & Jemal, 2020). Therefore, it is essential to identify novel targets for improving the clinical management of HCC.

S100 proteins are EF-hand calcium (Ca2+)-binding protein that play important roles in the progression, manifestation, and therapeutic aspects of a variety of diseases (Allgöwer et al., 2020). A previous study indicated that S100 proteins have tumorigenic gene functions and are dysregulated in multiple types of cancers, such as melanoma, gastric cancer, colon cancer, pancreatic cancer and leukemia (Allgöwer et al., 2020). S100 protein play a significant role in metastasis and establishment of a premetastatic niche in tumor microenvironment (Peinado et al., 2017). S100A4 has been shown to be secreted from stromal/tumor cells to release inflammatory chemokines such as CXCL8 and CCL2 in a paracrine manner, thereby promoting the metastasis of various cancers (Fei et al., 2017). The elevated S100A7 in mammary epithelial cells may enhance tumor growth through recruitment of macrophages to the tumors by activating RAGE and releasing proinflammatory cytokines (Padilla et al., 2017). S100A8/A9 recruits myeloid-derived suppressor cells through the activation of the serum amyloid A3 (SAA3)/TLR4 axis, which is crucial in establishing a pre-metastasis niche (Nasser et al., 2015).

S100B is an important member of the S100 family that is aberrantly expressed in various malignant tumors, including melanoma, breast cancer, colon cancer, metastatic lung cancer and ovarian cancer (Wu et al., 2020). At physiological levels, S100B regulates cancer cell proliferation and metabolism to modulate malignant tumor processes (Wu et al., 2020). Additionally, S100B can serve as a marker for metastasis in lung cancer, ovarian cancer, melanoma, and breast cancer. In lung cancer, S100B levels were shown to be elevated in patients with nonsmall cell lung cancer (NSCLC) with brain metastases (Chen et al., 2019). Moreover, S100B expression is correlated with metastatic potential and suppresses the migratory capacity of ER-negative breast cancer (Ertekin et al., 2020).

Hypoxia is known to play important roles in the development and progression of tumors. A recent study indicated that S100 family proteins are involved in the adaptive response to hypoxia (Hanze et al., 2003). It has been reported that hypoxia mimicked by cobalt chloride (CoCl2) enhances the mRNA and protein expression of the S100A4 gene in gastric cancer. Hypoxia-inducible factor 1 (HIF-1α) directly binds to the hypoxia response elements (HREs) in intron 1 of the S100A4 gene and results in its transcriptional activation (Horiuchi et al., 2012). Additionally, hypoxia and HIF-1α are also novel regulators of S100A8/A9 expression in prostate cancer (Grebhardt et al., 2012).

The S100 family is emerging as a novel diagnostic marker for identifying and monitoring various cancers. It has been reported that an S100B-effector protein interaction inhibitor, by interfering with the interactions with S100B and its effector protein, is a potential strategy to treat malignant tumors (Wu et al., 2020). However, studies of the association of S100B with liver cancer incidence have not been reported so far. Whether S100B is associated with the progression of HCC remains unknown. Therefore, the role of S100B in HCC progression, invasion, and metastasis needs to be explored in depth.

Materials and Methods

Cell culture

The human HCC cell line HepG2 and the immortalized human hepatic cell line HL-7702 (L02) were obtained from the Type Culture Collection of the Chinese Academy of Sciences (Shanghai, China). The cells were cultured with DMEM (Gibco, Waltham, MA, USA) including 10% fetal bovine serum (FBS) (Gibco, Waltham, MA, USA), 10 μg/mL streptomycin, and 10 μg/mL penicillin (Sigma, St. Louis, MO, USA). All cells were grown in an incubator at 37 °C in 5% CO2.

Plasmid construction and transfection

To construct a plasmid containing shRNA to inhibit target gene expression. The shRNA sequences of S100B and HIF-1α were obtained from online websites (https://rnaidesigner.thermofisher.com/rnaiexpress). Subsequently, the annealed oligonucleotides of shRNA primers were subcloned into the PGPU6/GFP/Neo vector (Gene Pharma, Shanghai, China). These plasmid constructions were synthesized by the gene pharma company. Plasmids expressing NC and shRNA were designated, shNC, shS100B, and shHIF-1α, respectively.

In cell transfection assays, when the cell density was reached to 70%, Lipofectamine 2000 (Invitrogen, Carlsbad, CA, USA)-mediated cell transfection was executed according to the manufacturer’s protocol. The Lipofectamine 2000 (5 μL) was diluted in 250 μL DMEM, and then mixed well with plasmid. Subsequently, the above solutions were added to the cell culture and placed in the incubator 5% CO2 at 37 °C. The efficiency of genetic silencing by the shRNA was evaluated by quantitative PCR (q-PCR).

The q-PCR analysis

RNA was extracted using TRIzol reagent (Invitrogen, Carlsbad, CA, USA) protocol. cDNA was obtained according to the protocol of the Prime Script™ RT Master Mix Kit (Takara, Kusatsu, Japan), and q-PCR assay was performed using SYBR Premix Ex Taq™II (Takara, Kusatsu, Japan) on a Thermal Cycler CFX6 System (Bio-Rad, Hercules, CA, USA). The relative gene expression levels were assessed using the 2−ΔΔCt method.

Western blot analysis

Total proteins were extracted from cells after transfection for 48 h using RIPA buffer with proteinase, and then BCA protein assay (Thermo Fisher Scientific, Waltham, MA, USA) were performed to assessed protein concentrations. The different concentrations of SDS-PAGE were used to separate target proteins. Subsequently, it was transferred on PVDF membranes and blocked with 5% nonfat milk. Protein-bound membranes were incubated with primary and secondary antibodies, respectively. The ECL substrates was used to detect the signals and photo record (Millipore, Burlington, MA, USA).

Hypoxia response element

We obtained the genomic sequence of human S100B from NCBI and screened for the presence of HREs 2,000 bp upstream of the TSS sequence. The HRE sequence was “(A/G)CGT(G/C)”.

Luciferase reporter assay

The luciferase reporter assay related plasmids were transfected into HepG2 cells. Subsequently, the cells were washed with PBS and treated using the Luciferase Reporter Assays Substrate Kit (ab228530; Abcam, Cambridge, MA, USA). Then, the Dual Glo Luciferase Assay System (Promega, Madison, WI, USA) and GloMax Multi Detection System (Promega, Madison, WI, USA) were used to confirm their luciferase activity. Their negative control was Renilla luciferase activity.

Chromatin immunoprecipitation (ChIP) assay

All collected cells were performed the assays according to the protocol from the Simple ChIP Enzymatic Chromatin IP Kit (Cell Signaling Technology, Danvers, MA, USA). DNA binding chromatin fragments were incubated with anti-HIF-1α antibodies and IgG (Sigma, St. Louis, MO, USA) and then the antibody and protein conjugate were immunoprecipitated with Protein G magnetic beads. Subsequently, the immunoprecipitated DNA was collected. The enrichment of particular DNA sequences was analyzed by q-PCR.

Cell viability assay (MTT assay)

Cell proliferation was measured by an MTT assay (Roche Diagnostics, Basel, Switzerland). The cells were digested and collected after transfection for 24, 48, 72, and 96 h, subsequently cultured in medium with 0.5 mg/ml MTS and kept in the dark for 4 h. Cell viability was assessed by the OD (Optical Density) value at 490 nm by an EnSpire Multimode Plate Reader (PerkinElmer, Suzhou, Japan).

Colony formation assay

The transfected cells were collected after digestion with trypsin. Then, the cells were seeded onto a six-well plate with a density of 1,000 cells/well. The cells grow in a humidified incubator (5% CO2, 37 °C). After routine incubation for 14 d, the colonies were stained with 1% crystal violet for 30 min, after which the number of colonies formed was counted.

Cell migration assay

After transfection for 24 h, the cells were scratched with 100 μL pipette tips, and lines were drawn in parallel. Subsequently, they were washed out with PBS buffer twice times. After 48 h of wound formation, the wound size was photographed and measured. The wound healing rate was calculated as follows: wound healing rate = [(scratch width at 0 h) − scratch width at 48 h/(scratch width at 0 h)] × 100%.

Trans well invasion assays

The cells were starved for 6 h in serum-free medium, trypsinized and adjusted 5 × 104 cells after counting. A total of 200 µL of the cell suspension was in the upper chamber, while 600 μL medium containing 10% serum was in the lower chambers. Then, the migrated cells in the lower chambers were fixed with 4% paraformaldehyde and stained with 0.5% crystal violet. The stained cells were observed under a Carl Zeiss microscope system (ZEISS, Jena, Germany) and recorded cell numbers after 24 h later.

S100B expression-related genes analysis in LIHC

The S100B expression related genes were obtained using the LinkedOmics based on TCGA_LIHC cancer cohort (HiSeq RNA platform) data. GO (Gene Ontology) according to biological processes, and KEGG (Kyoto Encyclopedia Genes and Genomes) analysis based on gene set enrichment analysis was performed in TCGA_LIHC. In addition, the relationships between S100B expression and cancer pathways activation in LIHC were assessed via GSCA database (Liu et al., 2018a).

Construction of PPI (protein–protein interaction) networks

The STRING database was used to analyze PPI networks between S100B and EMT-related genes. The networks were constructed by Cytoscape version 3.8.0. Furthermore, S100B directly interacting proteins were predicted using GeneMANIA.

The immune cell infiltration analysis

HPA (Human Protein Atlas) was used to analyze S100B level in normal live cells based on single cell sequencing results. Genes analyze module of CancerSCEM (Cancer Single-cell Expression Map) was used to confirm the expression of S100B in different cell subtypes based on single cell RNA-Seq datasets. S100B expression related tumor-infiltrating immune cell signatures in LIHC (Liver Hepatocellular Carcinoma) was investigated through TIMER (Tumor IMmune Estimation Resource) and GSCA (Gene Set Cancer Analysis) database (Li et al., 2017). The relationship between S100B expression and immune-related molecules were explored using TISIDB (Tumor-Immune System Interactions) portal based on the TCGA data (http://cis.hku.hk/TISIDB/) (Ru et al., 2019). Potential effects of S100B expression and immune cell infiltration level on overall survival (OS) were evaluated in TIMER2.0 via Kaplan–Meier plotter in HCC.

Statistical analysis

All experiments were independently performed at least three times. Their results were expressed as the mean ± SD. Two-tailed Student’s t-test was performed for comparing the data between two groups. Correlations were analyzed using Pearson’s correlation coefficient analysis. Differences were considered as statistically significant P values < 0.05.

Results

S100B-related functions enrichment analysis

To confirmed the mechanism of S100B during HCC progression, we firstly explored its functions through online tool TISIDB. As shown in Table 1, biological processes, such as response to hypoxia and I-kappaB kinase/NF-kappaB signaling, were remarkably regulated by S100B. In molecular function, S100B was involved in protein binding, calcium ion binding and RAGE receptor binding. In addition, pathways, including TLR4 signaling, glycosylation endproduct receptor signaling and immune related pathways, were found to be associated with S100B alterations. Therefore, we focused on the functions of S100B in hypoxia responses and immune related signaling pathways in liver cancer.

Table 1 GO functional enrichment analysis predicted three main functions of S100B, including biological process, cellular components and molecular functions.

Biological process	GO:0001666 response to hypoxia	
GO:0007249 I-kappaB kinase/NF-kappaB signaling	
GO:0007409 axonogenesis	
GO:0007517 muscle organ development	
GO:0007519 skeletal muscle tissue development	
GO:0007611 learning or memory	
GO:0007613 memory	
GO:0008360 regulation of cell shape	
GO:0010001 glial cell differentiation	
GO:0014706 striated muscle tissue development	
Molecular function	GO:0044548 S100 protein binding	
GO:0048156 tau protein binding	
GO:0048306 calcium-dependent protein binding	
GO:0050786 RAGE receptor binding	
Cellular component	GO:0001726 ruffle	
GO:0031252 cell leading edge	
GO:0043025 neuronal cell body	
GO:0044297 cell body	
Reactome pathway	R-HSA-166054: Activated TLR4 signaling	
R-HSA-879415: Advanced glycosylation endproduct receptor signaling	
R-HSA-168256: Immune system	
R-HSA-168249: Innate immune system	
R-HSA-1834949: Cytosolic sensors of pathogen-associated DNA	
R-HSA-3134963: DEx/H-box helicases activate type I IFN and inflammatory cytokines production	
R-HSA-975871: MyD88 cascade initiated on plasma membrane	
R-HSA-975155: MyD88 dependent cascade initiated on endosome	
R-HSA-166166: MyD88-independent TLR3/TLR4 cascade	

Hypoxia enhances S100B expression via HIF-1α signaling in HCC

To further confirm the function of S100B in hypoxia biological process, the expression of S100B under hypoxic conditions was assessed. We found that S100B was significantly upregulated in HepG2 cells under hypoxic conditions (Figs. 1A, 1B). It is well known that HIF-1 is a master regulator of the hypoxia response in many cancers. We also fund that HIF-1α was upregulated in HepG2 cells under hypoxic conditions (Fig. 1C). To assess whether HIF-1α is a driver factor of S100B upregulation in hypoxia, we constructed the shHIF-1α to knock down its expression in HepG2 cell (Fig. 1D). Furthermore, hypoxia-induced S100B expression could be abolished after knockdown of HIF-1α in HepG2 cells under hypoxic conditions (Figs. 1E, 1F). Therefore, 100B is a direct transcriptional target of HIF-1α in HCC.

Figure 1 S100B is induced by hypoxia in HepG2.

(A, B) The mRNA and protein level of S100B in HepG2 cells under hypoxic conditions. (C) HIF-1α mRNA levels in HepG2 cells under hypoxic conditions. (D) Knockdown of HIF-1α by shRNA in HepG2 cells. (E, F) The mRNA and protein level of S100B and HIF-1α in HIF-1α downregulated HepG2 cells under hypoxic conditions. NC, negative control; shHIF-1a, shRNA targeting HIF-1a mRNA. GAPDH was used to standardize HepG2 cells S100B expression in q-PCR and western blotting experiments. *P < 0.05, **P < 0.01, ***P < 0.001, ****P < 0.0001.

Since HIF-1α binds to HREs in the promoters of hypoxia-responsive genes to induce their transcription, we analyzed the potential HREs in the S100B promoter. Two potential HREs were found in the S100B promoter (Fig. 2A). According to the HREs locations, three S100B promoter expression vectors, including promoter 1 containing two HREs and two truncated mutations, promoter 2 containing one HREs and promoter 3 no HREs were constructed and the dual luciferase reporter assays were performed. As expected, HIF-1α overexpression dramatically enhanced S100B promoter activity but did not affect its activity of truncated promoter with deletion of two HREs (Fig. 2B). In addition, ChIP assays also confirmed that HIF-1α bound to the HREs of the S100B promoter. Moreover, HIF-1α also enriched in the S100B promoter under hypoxic conditions (Fig. 2C). Further regression analysis using cBioPortal revealed that S100B mRNA level was positively correlated with HIF-1α expression in liver cancer (Fig. 2D).

Figure 2 S100B is a direct target of HIF-1α in hypoxia.

(A) The potential HREs of the S100B promoter in the JASPAR database (http://jaspar.genereg.net/). Two green ovals represent the positions of HREs in the S100B promoter. (B) Relative activities of the S100B promoter and HREs mutants assessed by the dual luciferase reporter assays in HepG2 cells. Promoter1 (−980 bp) includes two HREs motifs; Promoter2 (−844 bp) contains one HRE motifs; Promoter3 (−726 bp) does not contain any HREs; Promoter (NC) is empty expression vector as a negative control. ****P < 0.0001. (C) Assessment of HIF-1α in the S100B promoter assessed by ChIP under normoxic and hypoxic conditions. The antibody is HIF-1α, a negative control is IgG. q-PCR was used to analyze the crosslinking status of HIF-1α in the S100B promoter. *P < 0.05, ***P < 0.001. (D) Regression analysis between S100B and HIF-1α in liver cancer using cBioPortal.

Hypoxia-related HepG2 cell migration and invasion partially depend on S100B

It is well known that the hypoxic condition drives EMT leading to aggravate cancer progression in solid tumors. We analyzed the biological functions of S100B in HepG2 cells under hypoxic conditions. A wound healing assays showed that the migratory ability of HepG2 cells was significantly increased under hypoxic conditions (Figs. 3A, 3B). We then assessed the migratory ability of HepG2 cells with S100B knockdown (Figs. S1A, S1B). As expected, the migratory rate was significantly hindered following downregulation of S100B, which is similar to the inhibitory effect observed in HIF-1α downregulated HepG2 cells (Figs. 3C, 3D). Consistent with the cell migration results, the invasion assays also showed that hypoxia’s promoting effect was eliminated by HIF-1α and S100B knockdown (Figs. 3E–3H). These results indicated that knockdown of S100B may inhibit hypoxia-induced HepG2 cell migration and invasion.

Figure 3 Hypoxia-induced HepG2 cell migration and invasion depend on S100B.

(A, B) The migration ability of HepG2 cells under hypoxic conditions. (C, D) HIF-1α and S100B knockdown represses the migration ability of HepG2 cells under hypoxic conditions. (E–H) Hypoxia promotes HepG2 cell invasion (E, F), while HIF-1α and S100B knockdown represses the invasion capability of HepG2 cells under hypoxic conditions (G, H). NC indicates negative control; shHIF-1a represents shRNA targeting HIF-1a mRNA; shS100B means shRNA targeting S100B mRNA. *P < 0.05, ***P < 0.001, ****P < 0.0001.

The enrichment analysis and co-expression profiles of S100B in HCC

We further investigated the functions of S100B and its co-expressed genes using GSVA enrichment analysis in HCC. We found that S100B expression was positively correlated with epithelial mesenchymal transition (EMT) and DNA Damage activity, and negatively correlated with TSC mTOR activity in LIHC (Fig. 4A). To confirm the functional downstream targets of S100B in HCC, the co-expressed genes of S100B were identified in LIHC. The results showed that 538 genes were positively correlated with S100B mRNA level (R > 0.5, P < 0.05), while no negatively correlated genes (R < −0.5 P < 0.05) were found in LIHC (Table S1). The top 50 genes that were significantly associated with S100B expression in LIHC were showed by heat map (Figs. S2A, S2B). Furthermore, the top 50 S100B-related genes were also positively associated with EMT activity in LIHC (Fig. 4A). Therefore, S100B is involved in EMT process of HCC cell.

Figure 4 The pathways and co-expression networks of S100B-related genes in HCC.

(A) The associations of S100B and top 50 S100B-related genes with the activity of cancer pathways in HCC. Purple represents positive correlation; green represents negative correlation. An asterisk (*) indicates P value ≤ 0.05; # represents FDR ≤ 0.05. (B) Venn diagram showing S100B-related genes in EMT of HCC. (C) PPI network between S100B and EMT-related proteins based on the results from STRING database. (D) S100B interacting proteins are predicted by GeneMANIA.

Subsequently, we obtained the hallmark EMT-related genes set from MSigDB. Veen analysis showed that eight EMT-related genes, such as BASP1, Col8a2, EMP3, ENO2, PLAUR, TGFB1, VIM, and WIPF1, were positively correlated with S100B expression in LIHC (Fig. 4B). Moreover, PPI network analysis results indicated that S100B only directly interacted with ENO2, and indirectly interacted with EMP3, VIM, and TGFB1 (Fig. 4C). Furthermore, the PPI network was also constructed via GeneMANIA. As shown in Fig. 4D, S100B had a positive correlation with 20 genes, such as CACYBP, CAPZA1/2, HMGB1, ANXA6, and other S100 members (S100A1, S100A11, S100P, S100A6, S100A12, and S100A4). Therefore, S100B likely promote cells invasion and migration to regulate these EMT-related genes in HCC progression.

S100B expression is involved in tumor immune infiltration of HCC

Function enrichment analysis of S100B associated gene sets were performed in HCC. We found that they were mainly involved in the biological processes, including adaptive immune response, leukocyte differentiation and activation, immune response-regulating signaling pathway, and response to chemokine (Fig. 5A). KEGG results indicated that these genes were primarily enriched in allograft rejection, Th17 cell differentiation, NF-kappa B pathway, T cells and B cells receptor, and chemokine signaling pathways in HCC (Fig. 5B). This result is consistent with the pathways analysis results through Reactome database, suggesting that S100B may be involved in regulating of inflammation and immune response in HCC.

Figure 5 S100B expression is associated with immune infiltration of HCC.

(A, B) GO and KEGG enrichment of S100B and functional partner genes in HCC. (A) S100B-related GO terms based on biological process annotations. (B) S100B-related KEGG pathways based on the GSEA analysis. Dark blue and orange indicates a false discovery rate ≤0.05; light blue and orange, a false discovery rate >0.05, respectively. (C) The correlations between the expression of S100B and the immune infiltration levels of six immune cell subtypes, including B cell, CD8+ T cell, CD4+ T cell, macrophage, neutrophil and DC cell in TIMER. (D) Correlations between S100B expression and immune infiltration levels of different immune cell subtypes in HCC. Pink represents positive correlation; blue represents negative correlation. An asterisk (*) indicates P value ≤ 0.05; # represents FDR ≤ 0.05. P < 0.05 is considered statistically significant.

Next, we investigated the relationship between S100B expression and infiltration levels of different immune cell subtypes in HCC. A significantly positive correlations were found between mRNA level of S100B and the degree of various immune cells infiltration in HCC. As shown in Fig. 5C, S100B was significantly correlated with four immune cells (cor > 0.45), including B cells, CD8+ T cells, macrophage cells, and myeloid dendritic cells. It was weakly associated with CD4+ T cells and neutrophil cells in HCC. Furthermore, we also investigated their correlations via the GSCA database. A positive relationship was found between S100B expression and the immune infiltration of CD8+ T cells, NK cells, DC cells and macrophage cells in HCC (Fig. 5D). Therefore, all the results suggested that S100B may be involved in immune cells infiltration of HCC.

S100B expression is involved in the functional network of immune molecules

We further investigated the relationship between S100B expression and immune molecules in HCC. S100B was positively associated with most of immune molecules in HCC. S100B expression had significant positive correlations with immuneostimulators in HCC (Figs. 6A, 6B). The immune-inhibitors, containing CTLA4 (R = 0.639), TIGIT (R = 0.638), and HAVCR2 (R = 0.631), and immune-stimulators, including CD48 (R = 0.693), CD86 (R = 0.673), and TNFRSFB (R = 0.66), were positively correlated with S100B expression, respectively (Figs. S3A, S3B). Moreover, it was closely associated with MHC genes (Fig. 6C), especially HLA-DPB1 (R = 0.623), HLA-DOA (R = 0.62), and HLA-DPA1 (R = 0.593) (Fig. S3C). Additionally, the positive relationships also were found with many chemokines and all listed chemokine receptors (Figs. 6D, 6E). The top three chemokines included CCL5 (R = 0.598), CCL22 (R = 0.59), and CCL4 (R = 0.57), and the top three chemokines’ receptors were CCR5 (R = 0.652), CXCR3 (R = 0.598), and CXCR4 (R = 0.591) (Figs. S3D, S3E). Additionally, the correlation of S100B expression and immune checkpoints, such as PDCD-1, CTLA4, HAVCR2, and TIGIT were confirmed in HCC. The results showed that expression of these genes was significantly correlated with S100B expression in HCC (Figs. 6F–6I). Thus, S100B expression maybe result in the migration of immune cells in HCC tumor microenvironment.

Figure 6 Correlation analysis of S100B expression with immunomodulators, chemokines/chemokine and immune checkpoints in HCC.

(A, B) The correlation between S100B and immunomodulators, immune-inhibitors (A), and immune-stimulators (B). (C) The correlation between S100B and MHCs. (D, E) The correlation of S100B with chemokines (D) and chemokine receptors (E) in HCC. Purple represents positive correlation; green represents negative correlation. (F–I) The relationship between S100B and immune checkpoints, including PDCD1 (F), CTLA4 (G), HAVCR2 (H), and TIGIT (I) in HCC.

S100B expresses in immune cells at the tumor tissues of HCC

Hypoxia is a common characteristic of solid tumors. To further explore S100B expression in immune cells at the hypoxia microenvironment, we assessed its expression in immune cells based on the single cell sequencing results. HPA database showed that S100B was mainly centralized in T cells and Kupffer cells in normal liver (Figs. 7A, 7B). Subsequently, the expression of S100B in different cell subtypes was further assessed via CancerSCEM database based on the single-cell RNA-Seq results in HCC. We found that S100B was observed in the immune cell at the tumor tissues, especially NK cell in HCC (Figs. 7C, 7D).

Figure 7 The expression of S100B in immune cells based on the single-cell sequencing results.

(A, B) S100B is majorly expressed in T cells and Kupffer cells inliver. (A) The UMAP dimensionality reduction results of scRNA-seq results from HPA database. (B) The expressions of S100B in liver cells is shown by histogram. (C, D) S100B is observed in NK cells at the tumor tissues of liver cancer. (C) Whole expression profiles of S100B in single-cell sample of liver cancer from CancerSCEM database. (D) The expressions of S100B in different cell subtypes in the HCC sample. (E–I) The OS survival analysis of the S100B expression and immune cells infiltration levels in HCC. Kaplan-Meier curves of OS stratified by S100B expression in combination with immune cells, including CD8 T (E), T cell CD4+ Th2 (F), macrophage (G), neutrophil (H), DC (I) in HCC.

In addition, the prognosis of patients with HCC was investigated based on S100B mRNA expression and different immune cells infiltration levels. The OS outcome of low S100B/ high CD8+ patients were significantly better than other groups (Fig. 7E). While higher infiltration levels of CD4+Th2, macrophage, neutrophil, and DC cells with high S100B expression were connected with shorter OS (Figs. 7F–7I). However, there was no significant different was found in the group with B cells infiltration and S100B expression. Therefore, these results implicated that S100B may affect the prognosis of patients with HCC in part due to immune cells infiltration.

The hub genes of S100B co-expression and hypoxia response in HepG2 cell are associated with immune infiltration

To validate how up-regulation of S100B by hypoxia might be correlated to the immune cell infiltration, we performed an association analysis between the S100B significantly co-expressed genes and the differential expression profiles of HepG2 cells in hypoxia response based on the microarray data of GSE15366. Venn diagrams showed that 53 genes were differentially expressed in the hypoxia and associated with S100B expression in HCC (Fig. 8A). These genes were showed in Table S2. Furthermore, enrichment analysis results revealed that these genes were strongly associated with immune-related terms/pathways, such as regulation of T cell activation, adaptive immune response, modulators of TCR signaling and T cell activation, mast cell activation, and neutrophil degranulation (Fig. 8B). Moreover, we found that the expression of these gene sets has a closely association with tumor immune cell infiltration (Cor = 0.78), including Tfh, Th1, CD8 T cell, exhausted cell, NK cell, macrophage, DC, and CD4 T cell (Fig. 8C). Due to hypoxia enhances S100B expression via HIF-1α signaling in HCC, HIF-1α-S100B axis related differentially expressed genes were identified in HCC. As shown in Table S2, 34 genes were obtained in two groups. They also involved in regulation of immune response. Next, we used the GSCA database to analyze the correlation between their expression and immune infiltrating cells in LIHC. The results showed that the expression of these genes was positively associated with the degree of infiltration of immune cells (Cor = 0.59), including central memory, macrophage, Tfh, CD4 T cell, DC and iTreg (Fig. 8D). Therefore, these results suggested that up-regulation of S100B by hypoxia might regulate these genes expression to affect the immune cell infiltration of LIHC.

Figure 8 Hub genes of S100B co-expression network in hypoxia are associated with immune infiltration of HCC.

(A) Venn diagram show the share genes from groups of S100B co-expressed genes set, differentially expressed genes of hypoxia response in HepG2, and HIF-1α co-expressed genes set in LIHC. (B) The functional enrichment analysis of these hub genes between S100B co-expressed genes and hypoxia response genes in HepG2. (C) These hub genes expression is correlated with immune infiltration level in HCC. Pink represents positive correlation; blue represents negative correlation. (D) The expression of hub genes between S100B and HIF-1α co-expressed gene set are associated with immune infiltration level in HCC. Orange represents positive correlation; blue represents negative correlation. An asterisk (*) indicates P value ≤ 0.05; # represents FDR ≤ 0.05. P < 0.05 is considered statistically significant.

Discussion

Due to the rapid progress of liver cancer and few effective drugs for therapy, new therapeutic targets for the improved prognosis have been the focus of recent liver cancer studies. Numerous studies have reported that S100B is involved in tumor metastasis in many cancers (Seguella et al., 2019). It is a well-established biomarker for the diagnosis and staging of multiple tumors in a clinical setting. Moreover, S100B inhibitors have been reported to target the S100B-p53 interaction, which is beneficial for melanoma therapy (Wu et al., 2020). However, the function of S100B in HCC remains unclear.

Hypoxia is a prominent characteristic of solid tumors, which plays an important role in tumor angiogenesis, cell proliferation, stemness maintenance, cell differentiation, apoptosis in HCC (Graham & Unger, 2018; Ling et al., 2020). Although several hypoxia-related genes have been identified to be closely associated with cellular processes in HCC, the mechanism by how this occurs is very limited. Our study shows for the first time that S100B is significantly upregulated under hypoxic conditions. Moreover, knockdown of S100B represses hypoxia-induced cells migration and invasion ability of HepG2 cell. More importantly, we confirm that S100B is likely a functional downstream target of HIF-1α for cell metastasis in HCC. HIF-1α directly binds with the HREs of the S100B promoter to induce its activation. It is consistent with the previous results that HIF-1α transactivates VASP expression through HRE binding, which in turn dysregulates the actin cytoskeleton to promote invasion and metastasis in HCC (Liu et al., 2018b). Additionally, hypoxia is aggressively conducive to metastatic states in many cancers. We demonstrate that S100B and its co-expressed genes are primarily involved in EMT of HCC. S100B directly interacts with CAPZA1, which has been reported to drive EMT through binding to F-actin in hypoxia, thereby inducing actin cytoskeleton remodeling in HCC (Huang et al., 2019). Taken together, S100B is critical for hypoxia-induced EMT in HCC.

In the current study, GO and pathways enrichment analysis results indicate that S100B and its co-expressed genes are involved in tumor-related immune response pathways, including NF-kappa B signaling, T cell and B cell receptor signaling pathway in human cancer. Moreover, S100B expression is significantly correlated with immune cells infiltration, in particular for B cells, CD8+ T cells, macrophages, and dendritic cells. Thereby we speculate that S100B may be crucial for immune microenvironment in HCC. In recent years, many studies have confirmed that hypoxia is an early event in tumor evolution that has been shown to both directly and indirectly impact tumor immune microenvironment (Semenza, 2021; You et al., 2021). Hypoxia could increase the recruitment and/or polarization of immunosuppressive cell populations, and results in the resistance of tumor cells to immune attack and evade immunosurveillance (Palazon et al., 2014; Li et al., 2018). In this study, hypoxia and S100B related gene expression network analysis showed that hub genes shared between hypoxia related genes expression profiles of HepG2 cells and S100B co-expressed genes network have a crucial role in the immune infiltration, especially T cells, NK and macrophages. Additionally, the expression of downstream target genes of HIF-1α-S100B axis is also significantly associated with the immune cell infiltration in HCC. Therefore, these results indicate that hypoxia induced up-regulation of S100B is likely correlated with the immune cell infiltration by regulating these genes expression in HCC.

Hypoxia is a common characteristic of solid tumors, including HCC. Single-cell data could truly reflect the cellular composition of tumor tissue and gene expression signature in HCC. We found that S100B is expressed in immune cell, such as T cell and NK cell. Combined previous results that hypoxic TMEs and HIF-1α can directly affect the frequency of CD8+ T cells and NK cell function to result in immunosuppression due to a lack of cytotoxic cells (Vito, El-Sayes & Mossman, 2020; Balsamo et al., 2013). We speculate that hypoxia contributes to S100B overexpression in immune cells and promotes cells invasion and tumor immune escape through regulation of S100B related gene expression in HCC. However, this study only confirms the functions of S100B in vitro, the exploring the intricacies of the hypoxia-immune cell relationship in vivo has been a challenge. More in-depth studies are needed to reveal the role of S100B in the tumor microenvironment to be its more convincing in HCC.

Recently, it has been reported that immune checkpoint inhibitors have a remarkable clinical efficacy in various types of cancers (Zhang et al., 2017; Herbst et al., 2014). The anti-PD-1/anti-PD-L1-based combination therapy represents a promising strategy for HCC (Cheng et al., 2020). Hypoxia drives the immune-escape of tumors by inducing the expression of immune checkpoint inhibitors and controlling the antigen presenting mechanisms (Vito, El-Sayes & Mossman, 2020). In this study, we found that S100B expression is significantly associated with immune-related genes and immune checkpoints in HCC. Additionally, S100B interacting proteins, such as ICACYBP, HMGB1, and S100A8, have been reported to positively correlate with the response of immunotherapy in HCC (Peng et al., 2021; Li et al., 2021). Taken together, these findings validate the potential that combination therapy targeted S100B and immune checkpoints may be a strategy to improve the efficiency of immunotherapy in HCC.

Conclusions

Our study confirms that S100B maybe the key point to integrate hypoxia and immune responses in HCC microenvironment. S100B is activated by hypoxia via HIF-1a dependent manner to promoter cell proliferation, invasion and metastasis under tumor hypoxic conditions. Moreover, the signature of S100B could reflect the infiltration characterization of different immunocytes in the tumor microenvironment. In conclusion, our research gives new insights regarding the roles of S100B in HCC. It will be an effective therapeutic target to inhibit the invasion and migration driven by hypoxia in HCC cells. Moreover, combination of immunotherapies with S100B inhibitors may be helpful for the effective targeted therapy of liver cancer.

Supplemental Information

Supplemental Information 1 The knockdown of S100B in HepG2 cells.

(A, B) The shRNA targets S100B is transfected into HepG2 cells to downregulated S100B expression. (A) The fluorescence of cells after shRNA transfection in HepG2 cells. bar = 200 μm. (B) The mRNA level of S100B in shRNA transfected HepG2 cells. Data are shown as the mean ± standard deviation based on three independent experiments. **P < 0.01.

Click here for additional data file.

Supplemental Information 2 S100B co-expressed genes in LIHC.

(A, B) Heatmaps of the top 50 genes positively and negatively correlated with S100B, respectively. Orange represents positive correlation; blue represents negative correlation.

Click here for additional data file.

Supplemental Information 3 Correlation analysis of S100B expression with immunomodulators.

The top three immune-inhibitors (A) and immune-stimulators (B) associated with S100B expression in HCC. (C) The top three MHCs correlated with S100B expression in HCC. (D) The top three chemokines and (E) chemokine receptors involved in S100B expression in HCC.

Click here for additional data file.

Supplemental Information 4 S100B co-expressed genes in LIHC.

Click here for additional data file.

Supplemental Information 5 Hub genes between S100B co-expressed genes, hypoxia response genes, and SHIF-1α co-expressed genes in HCC.

Click here for additional data file.

Supplemental Information 6 Source raw data.

Click here for additional data file.

Supplemental Information 7 Raw data blots.

Click here for additional data file.

Additional Information and Declarations

Competing Interests

Author Contributions

Data Availability

The authors declare that they have no competing interests.

Jia Yan conceived and designed the experiments, analyzed the data, authored or reviewed drafts of the paper, and approved the final draft.

Ya jun Huang performed the experiments, prepared figures and/or tables, and approved the final draft.

Qing yu Huang performed the experiments, prepared figures and/or tables, and approved the final draft.

Peng Xia Liu analyzed the data, prepared figures and/or tables, financial support, and approved the final draft.

Chang Shan Wang analyzed the data, authored or reviewed drafts of the paper, and approved the final draft.

The following information was supplied regarding data availability:

The raw measurements are available in the Supplemental Files.

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
