# Peer review of "Comprehensive analysis of the correlations of S100B with hypoxia response and immune infiltration in hepatocellular carcinoma"

_PeerJ, doi:10.7717/peerj.13201_

## Round 0.1 · original submission · Major Revisions

Please address issues pointed out by the reviewers and amend the manuscript accordingly.

Reviewer 1 ·

Basic reporting

The English language needs improvement and mistakes corrected. Transfection and overexpression part in the methods as well as bioinformatic analyses and figure legends do are not provide sufficient information on what has been done.

Experimental design

The concept of the manuscript is unclear. The authors show hypoxic up-regulation of S100B expression and invasion in HCC cells and in HCC tumors to tumor cell migration. On the other hand, the subsequent analyses were performed with databases from HCCs and normal livers. As S100B appears to be mainly expressed in immune cells in normal liver (leaving cellular distribution in HCC unclear). It is unclear if hypoxia also up-regulates S100B expression in immune cells, how hypoxia is related to tumor hypoxia. In consequence, does this mean that S100B expression in HCC is due to increased immune cell infiltration in hypoxic HCC and immune cell infiltration is correlated with hypoxia and metastatic properties?
Moreover, this raises the question if the experimental investigations are relevant for second bioinformatics part of the manuscript.

Validity of the findings

The findings appear to be valid, but as the connection between the different parts of the manuscript and the use of the models is not convincing, the conclusions appear questionable.

Additional comments

1. Down-regulation of HIF-1alpha and S100B by shRNA expression plasmids in HepG2 cells is not an easy task considering the difficult transfection of these cells and the small efficacy of shRNAs. Neither plasmids are described, how they were generated nor the shRNAs nor is shown how effective the shRNA for S100B was.
2. Bioinformatic analyses are poorly described.
3. Figure legends are not instructive concerning the explanation how the data illustrated in the figures were generated.
4. What is “LIHC”?

·

Basic reporting

Very clear and organized reporting.

Experimental design

Experiments and analysis are well designed and executed.

Validity of the findings

Findings are valid, statistically sound and controlled. Conclusions are well stated.

Additional comments

Yan et al. reported that HIF-1α participates in the proliferation, invasion and metastasis of HCC through regulation of the expression of S100B in hypoxia. The study is well designed and analysis and conclusion well performed and written. The paper should be ready for publication after minor language/ presentation suggestions below are addressed:
- Figure 2A: There are two HREs (HRE1 and HRE2), but three promoter sites (-980, -844 and -726). Are the HREs in the three promoter sites? If they are, please label accordingly. Also, what does the two green oval represent? Please clarify in the description. Lastly, the HRE1 sequence (atgct) outlined does not have the consensus sequence of (A/G)CGT(G/C)
- Figure 2B: What is the role of the PGL3-basic? It is not mentioned anywhere in the text or Figure description
- Figure 4A: The blue and orange is not visible unless we zoom in a lot. Is it possible to make the coloring more obvious?
- Figure 5: Please put spaces between 'thatS100BmRNA' and 'normalhumanliver'. Also change 'it majorly expressed' to 'it is majorly expressed'. Please also provide clearer description of the takeaway for Figure 5A and 5B.
- Figure 6B and 10A: What does * and # signify?
- Figure 8: No H-J shown. Also, Figure 8G and 8D are the same.
- Line 14: qRCR. Shouldn't it be qPCR?
- Line 97: please delete the repeated 'Then,'
- Line 227: 'maintenance' can be changed to 'maintain'
- Line 285: 'bounds to' can be changed to 'binds'
- Line 319: 'S100B directly interacts proteins'. Does the author mean 'S100B-interacting proteins'?

Reviewer 3 ·

Basic reporting

Yan et al. used the Hepatocellular carcinoma (HCC) cell line HepG2 to investigate the role of S100B protein in progression, invasion, and metastasis.
The authors discovered that hypoxia increases S100B expression at both the mRNA and protein levels. They also discovered that HIF-1 binds to HREs in the S100B promoter region and increases its expression. They also discovered a direct association between S100B and HIF-1. In the HepG2 cell line, S100B is also involved in hypoxia-induced cell migration and invasion. The authors discovered that S100B was primarily found in T cells and Kupffer cells in normal liver, indicating a significant correlation with immune response in the liver. They discovered a positive relationship between S100B expression and infiltration levels of immune cells using TIMER analysis. They also reported a positive relationship with immunomodulators. Based on these findings, the authors claim that S100B expression aids in the recruitment and migration of immune cells in the HCC tumor microenvironment. Overall, this is a nice paper with carefully gathered data. It is well written and provides interesting mechanistic insights into how S100B regulates the HCC tumor environment under hypoxic conditions. S100B may also serve as a biomarker and drug target in the treatment of liver cancer. This manuscript, in my opinion, is appropriate for the PeerJ journal.

Experimental design

NA

Validity of the findings

NA

Annotated reviews are not available for download in order to protect the identity of reviewers who chose to remain anonymous.

---

## Round 0.2 · Major Revisions

Although reviewer #1 raised serious concerns and recommended rejection, I have decided to give you another opportunity to address corresponding critiques and amend the manuscript accordingly.

Reviewer 1 ·

Basic reporting

The manuscript is somehow improved, but not all and even my key concerns have not been addressed adequately. The authors show experimentally that hypoxia regulates S100B expression in tumor cells, but S100B appears to enter tumors mainly in the form of immune cells, Thus, the hypoxic regulation of immune cells appears to mediate hypoxia-induced upregulation of S100B in HCC. The authors still did not link their findings in HCC cells with their data base analyses.

Experimental design

The data base and the experimental part are not convincingly linked.

Validity of the findings

The validity of the findings is compromised by the study design

Additional comments

If the authors find a better way how up-reguation of S100B by hypoxia in the cells might be related to the immune cell infiltration and S100B in immune cells, one could re-consider the manuscript. Hypoxic regulation of S100B expression by HIF1alpha, regulating migration is interesting.

·

Basic reporting

Reporting is clear and unambiguous

Experimental design

The experimental design is well defined and methods sufficiently described

Validity of the findings

The findings and conclusions are well supported

Additional comments

My concerns have been mostly been addressed in the edited manuscript. There are only a few minor suggestions for change:
- Line 16: please define what NK cell is
- Line 24: please change ChIP-qRCR to ChiP-qPCR

---

## Round 0.3 · accepted · Accept

Thank you for addressing the remaining concerns and for revising your manuscript.